# Investigation of an Impedimetric LaSrMnO_3_-Au/Y_2_O_3_-ZrO_2_-Al_2_O_3_ Composite NO_x_ Sensor

**DOI:** 10.3390/ma15031165

**Published:** 2022-02-02

**Authors:** Nabamita Pal, Gaurab Dutta, Khawlah Kharashi, Erica P. Murray

**Affiliations:** Institute for Micromanufacturing, Louisiana Tech University, Ruston, LA 71272, USA; omsoosoo2010@hotmail.com

**Keywords:** composite electrolyte, NO_x_ sensor, impedimetric gas sensing, LaSrMnO_3_-Au composites, dense electrode, porous electrolyte, partially stabilized zirconia, fully stabilized zirconia

## Abstract

Composite NO_x_ sensors were fabricated by combining partially and fully stabilized yttria-doped zirconia with alumina forming a composite electrolyte, Y_2_O_3_-ZrO_2_-Al_2_O_3_, and strontium-doped lanthanum manganese oxide mixed with gold to form the composite sensing electrode, La_0.8_ Sr_0.2_MnO_3_-Au. A surface chemistry analysis of the composite sensor was conducted to interpret defects and the structural phases present at the Y_2_O_3_-ZrO_2_-Al_2_O_3_ electrolyte, as well as the charge conduction mechanism at the LaSrMnO_3_-Au electrode surface. Based on the surface chemistry analysis, ionic and electronic transport properties, and microstructural features of sensor components, the working principle was described for NO_x_ sensing at the composite sensor. The role of the composite materials on the NO_x_ sensing response, cross-sensitivity to O_2_, H_2_O, CO, CO_2_, and CH_4_, and the response/recovery rates relative to sensor accuracy were characterized by operating the composite ${\mathrm{NO}}_{x}$ sensors via the impedimetric method. The composite sensors were operated at temperatures ranging from 575 to 675 °C in dry and humidified gas environments with NO and NO_2_ concentrations varying from 0 to 100 ppm, where the balance gas was N_2_. It was found that the microstructure of the composite NO_x_ sensor electrolyte and sensing electrode had a significant effect on interfacial reactions at the triple phase boundary, as well as the density of active sites for oxygen reactions. Overall, the composite NO_x_ sensor microstructure enabled a high NO_x_ sensing response, along with low cross-sensitivity to O_2_, CO, CO_2_, and CH_4_, and promoted NO detection down to 2 ppm.

## 1. Introduction

As emission laws become more stringent, modern diesel engines are becoming more environmentally friendly, while offering greater torque, increased durability, higher fuel economy, and lower CO_2_ emissions than their predecessors. Additionally, while the range capacity, battery lifetime, and availability of nationwide charging stations currently limit electric vehicle feasibility for long-distance commercial trucking operations, the demand for diesel engines continues. However, despite the benefits of modern diesel engines, the exhaust generated contains nitric oxides (i.e., NO_x_) that contribute to air pollution. Efforts to substantially reduce NO_x_ emissions are necessitated by the push toward near-zero emissions by regulatory agencies in the United States and other countries [1,2,3,4,5]. Progress in diesel exhaust remediation has created the need for higher-accuracy NO_x_ sensors for onboard diagnostic systems to monitor and regulate diesel engine operation. Key factors that restrict higher-accuracy NO_x_ sensing include limited sensitivity to the analyte gas, cross-sensitivity to other exhaust gases, and sluggish sensor response and recovery rates. Conventional NO_x_ sensors are based on single-phase materials for the electrolyte and sensing electrode. However, composite materials for NO_x_ sensing components are attractive for promoting higher-accuracy NO_x_ sensing, as specific sensing characteristics can be tuned based on the composition of the composite electrolyte and composite sensing electrode to target multiple key factors governing the sensor response toward the analyte gas.

The sensing behavior of various metals and metal oxides has been studied in exhaust gas environments for potential composite electrode materials for NO_x_ sensors [6,7,8,9,10,11,12,13,14]. The resistance of the nanostructured grain–grain boundaries of the metal oxide change as a function of the NO_x_ gas upon surface adsorption of NO_x_ molecules, which influences the depletion layer and potential barrier. Thus, the overall electrical conductivity enhances the overall NO_x_ sensitivity [9,14]. Some of these studies have found that combining a noble metal with a metal oxide can enhance the sensing response to the analyte gas. Furthermore, high sensitivity to NO_x_ was achieved by varying the noble metal to metal oxide ratio composition, which was found to tailor the sensor response while achieving significantly less cross-sensitivity from interfering gases, thereby improving sensor accuracy for analyte gas [10,14,15]. For example, in work by Romanytsia et al., it was reported that composite sensing electrodes containing Au with 10 wt% yttria-stabilized zirconia (YSZ) produced significantly higher sensitivity to NO_2_ with a more rapid response rate for NO_2_ concentrations as low as 20 ppm, in comparison to NO_x_ sensors, which were utilizing pure Au sensing electrodes [12]. Studies on La-based perovskites indicated that LaSrMnO_3_ (LSM) caused the sensing electrode to become extremely sensitive to NO_x_ due to the Sr component [16]. Pal et al. later demonstrated that adding 10 wt% gold to LSM to form an LSM-Au composite sensing electrode promoted NO sensitivity down to 5 ppm and cross-sensitivity to H_2_O, CO, CO_2_, and CH_4_ was substantially reduced [10]. Another benefit of the metal oxide composite electrode was greater fabrication compatibility. Since Au has a relatively low melting temperature (~1060 °C), incorporating Au with the metal oxide resulted in a composite electrode that was more tolerant to the high-temperature firing processes generally associated with solid-state gas sensor fabrication. Moreover, the metal oxide component of the composite aided the stability of the Au particles within the sensing electrode during sensor operation.

Although composite electrolytes have been considered for other devices, such as solid oxide fuel cells, they have received less attention for solid-state NO_x_ sensor applications [17,18,19]. Composite electrolytes can increase the mechanical integrity of the device, as well as improve the ionic conductivity through the bulk material and along grain boundaries. Among the limited studies relevant to sensing NO_x,_ it has been reported that adding low amounts (i.e., 0.5–2 wt%) of Al_2_O_3_ to YSZ removes SiO_2_ impurities from grain boundaries, thereby facilitating ionic conductivity along the grain boundaries [20,21]. Porous composite electrolyte supported NO_x_ sensors evaluated by Kharashi et al. found that adding 2 wt% Al_2_O_3_ to the partially stabilized zirconia (PSZ) electrolyte promoted greater NO_x_ sensitivity compared to the non-composite PSZ supported sensors [22]. Increasing the Al_2_O_3_ concentration beyond 2 wt% adversely impacted sensor sensitivity to NO as the insulating properties of Al_2_O_3_ became apparent [23]. Related composite electrolyte studies suggested that the presence of PSZ in the sensor electrolyte may be beneficial for reducing water cross-sensitivity, thereby promoting NO_x_ sensor accuracy [24].

The present study explores the influence of composite electrode and composite electrolyte materials on the impedimetric response of NO_x_ sensors to further NO_x_ gas-sensing capabilities in the single ppm detection range. LaSrMnO_3_-Au was selected as the composite sensing electrode, since these materials have demonstrated high sensitivity and selectivity to NO_x_ [10,12,16]. The composite NO_x_ sensors included a porous composite electrolyte composed of fully stabilized and partially stabilized ZrO_2_ (i.e., FSZ and PSZ) along with Al_2_O_3_. The authors selected these materials, as prior studies suggested that composite electrolytes contributed to greater NO_x_ sensing capabilities, along with reduced water cross-sensitivity [22,24]. To the authors’ knowledge, other studies have not evaluated the behavior of NO_x_ sensors entirely composed of composite materials. The electrochemical behavior, NO_x_ sensitivity, cross-sensitivity to interfering gases, and the gas sensing response/recovery rates are discussed relative to the composite microstructures, defect formation, and structural phase influencing the working mechanism of the composite NO_x_ sensor.

## 2. Experimental

The composite NO_x_ sensors comprised an electrode support composed of 90 wt% La_0.8_Sr_0.2_MnO_3_ (LSM–Inframat Advanced Materials, Manchester, CT, USA) and 10 wt% gold (Au–Alfa, Aesar Haverhill, MA, USA). The electrode powders were dry mixed, pressed into pellets, and fired at 1400 °C for 1 h. Further details concerning electrode processing are discussed elsewhere [10]. The resulting electrodes were disc-shaped with a diameter of 11 mm and a thickness of about 1.1 mm (Figure 1). Archimedes measurements performed on LSM-Au electrode pellets indicated the electrodes tended to be 93 ± 2% dense. Fully stabilized zirconia (FSZ, 8 mol% Y_2_O_3_-ZrO_2_, Tosoh Corporation, Tokyo, Japan), partially stabilized zirconia (PSZ, 4.7 mol% Y_2_O_3_-ZrO_2_, Advanced Ceramics, Tucson, AZ, USA), and alumina (α-Al_2_O_3_, Alfa Aesar) were used to fabricate the FSZ composite electrolytes. A slurry was made using a 50:50 vol% mixture of FSZ and PSZ ceramic powders, along with a 3 wt% polyvinyl butyral (Butvar B-76) binder and ethanol as the solvent. Moreover, 2 wt% Al_2_O_3_ added to the slurry was relative to the weight of the FSZ and PSZ powders. The composite electrolyte slurry was tumbled with zirconia media for 16 h. Several dense LSM-Au pellets were partially coated with the composite electrolyte slurry and fired at 1000 °C for 1 h to achieve a porous electrolyte microstructure. A porous counter electrode was fabricated by painting a slurry made from the LSM-Au powders onto the electrolyte surface. The counter electrode was approximately 2 mm in diameter. Following the counter electrode application, the samples were again fired at 1000 °C for 1 h. The resulting LSM-Au_dense_/FSZ-PSZ-Al_2_O_3_/LSM-Au_porous_ cells were identified as the FSZ composite sensors. For comparison purposes, LSM-Au_dense_/FSZ/LSM-Au_porous_ cells were fabricated and noted as the FSZ sensors.

The microstructure, morphology, and elemental analysis of the FSZ composite sensors and FSZ sensors were evaluated by using scanning electron microscopy (SEM) and energy-dispersive X-ray spectroscopy (EDS), using a Hitachi SU8230 and Hitachi FESEM S4800 (Hitachi, Chiyoda City, Tokyo, Japan). The crystalline phases of the NO_x_ sensors were determined via X-ray diffraction (XRD) analysis that was performed by using a Bruker D8 Discover system (Bruker, Billerica, MA, USA) with a scanning rate of 2°/min with 2θ varied from 10° to 85°, using CuKα radiation (λ = 1.54056 Å). The anode current and operation voltage during XRD was 40 mA and 40 kV, respectively. The surface analysis and chemical state of the FSZ composite sensor were evaluated by using X-ray photoelectron spectroscopy (XPS) performed with a Scienta Omicron ESCA 2SR XPS (Scienta Omicron, Taunusstein, Germany) at a chamber pressure of 3.6 × 10^−9^ mbar. During XPS, the pass energy for survey and region scans was 50 and 30 eV, respectively. The X-ray source in XPS was monochromated AlKα with an operating condition of 15 kV/450 W. The XPS calibration was performed with reference C1s 284.8 eV, and the measurement area was about 3 mm in diameter before sample evaluation. The peak fitting and deconvolution of obtained spectra in these samples was performed by using CasaXPS software (CASA Software Ltd., Cheshire, UK) with a line shape: GL (30) (Gaussian/Lorentzian product formula, where the mixing was 30/100 = 30%, peak fitting Gaussian/Lorentzian ~30/70). No Argon sputtering was used during XPS measurements.

In order to electrochemically characterize the NO_x_ response, the sensors were placed in a quartz tube, loaded into a tube furnace, and operated at temperatures ranging from 575 to 675 °C. The gases exposed to the sensors contained NO and NO_2_ concentrations varying from 0 to 100 ppm, where O_2_ concentrations were 5–18 vol% in dry and humidified (10 vol% H_2_O) gas conditions with N_2_ as the balance gas. The variation in O_2_ and the addition of H_2_O was carried out to evaluate cross-sensitivity. The interference of CH_4_, CO_2_, and CO gases found in diesel exhaust was also tested. Gas-flow rates of 100 sccm were maintained by using a standard gas handing system. Electrochemical characterization of the NO_x_ sensors was carried out by using a Gamry Reference 600 (Gamry, Warminster, PA, USA). Impedimetric operation of the sensors was performed by applying a signal amplitude of 100 mV over a frequency range of 1 Hz–1 MHz and measuring the electrical response. Impedimetric measurements were taken in triplicate to ensure that the data were stable and reproducible. The equivalent circuit fitting was performed by EC-lab Software V10.40 (BioLogic, Seyssinet-Pariset, France) via the nonlinear least-squares method, with circuit parameters and error range listed in Section 3.2.3. The error values associated with circuit parameters in the table in Section 3.2.3 were the standard deviation calculated by the EC-lab software V10.40. Origin Pro software (ver. 2021b) (OriginLab Corp, Northampton, MA, USA) was used for the statistical calculations as well as graphical presentation of the experimental data.

## 3. Results and Discussion

### 3.1. Microstructure and Morphology of NO_x_ Sensors

#### 3.1.1. Surface Structure and Elemental Mapping via SEM and EDS

The comparisons of the SEM images shown in Figure 2a,b for typical FSZ and FSZ composite electrolytes indicated that the FSZ electrolyte possessed a fine and relatively uniform microstructure. In contrast, the FSZ composite electrolyte was composed of both coarse and fine irregularly shaped particles. The SEM images of the FSZ composite electrolyte cross-section are shown in Figure 2c–f, along with corresponding elemental mapping by EDS. The microstructure shown in Figure 2g–i corresponds to the cross-section of LSM-Au pellets and includes backscattered imaging in Figure 2g. Elemental mapping verified that the circular particles were Au (see Figure 2i). The EDS analysis exposed the presence of various elemental phases of the LSM-Au electrode. Appendix A (details in Appendix A Section S1) shows that the Sr doping level was ~0.22 mole and La ~0.44 mole, which indicated a lack of La/Sr ordering consistency. The oxygen excess nonstoichiometric isotropic properties at the LSM-Au surface benefit charge transfer [25,26]. The EDS analysis showed (Appendix A) Sr/La ~0.5, indicating less propensity of Sr depletion, which can minimize the formation of resistive SrZrO_3_ phases. Less formation of such a resistive phase was consistent with other reported findings (Appendix A Section S1) [26,27]. The perovskite La_(1−x)_Sr_x_MnO_3_ with a Sr doping level (x) was approximately ≥0.15 mole, resulting in hole hopping along the Mn sublattice. Such a scenario happens results from an energetically favorable double-exchange mechanism [25].

#### 3.1.2. Defect Estimation and Structural Phase Behavior via XRD

The XRD analysis in Figure 3 revealed that the FSZ composite electrolyte contained cubic phases and trace amounts of monoclinic and tetragonal phases. The combination of tetragonal-cubic phases in the FSZ composite was advantageous for stability against aging [28]. The tetragonal grains doped with yttria showed a heterogeneity match with the cubic phase that maintained a unimodal microstructure [28]. Due to the higher mol% of yttria in FSZ, the monoclinic peaks associated with PSZ became negligible in the FSZ composite electrolyte, and peak positions of tetragonal phases shifted toward more distorted cubic phases (see Figure 3b,c) [29]. Thus, PSZ with ~5 mol% yttria presented less volumetric monoclinic content than other PSZ with comparatively lower mol% of yttria [28,30].

The intrinsic stable defect formation at cubic phases of FSZ was mostly of Schottky type [31]. The porous FSZ composite electrolyte possessed Y_Zr_^′^, which was the Y^3+^ impurity in the Zr^4+^ site in the FSZ composite lattice [32]. The cubic phases in the FSZ composite had maximum Y_Zr_^′^-Y_Zr_^′^ interactions, with Y atoms occupying sites near the free lattice oxygen vacancy, V_Ö_^″^ (i.e., less Y_Zr_^′^-V_Ö_^″^ separations within the Y_Zr_^′^-V_Ö_^″^-Y_Zr_^′^ defect clusters) [33]. The XRD data indicated that the FSZ composite contained structural phases that favored oxygen ion transport on aliovalent Y doping [33]. The FSZ non-composite electrolyte had more extrinsic oxygen vacancies in defect clusters Y_Zr_^′^-V_Ö_^″^-Y_Zr_^′^. Compared to the FSZ composite electrolyte, the defect clusters in the FSZ non-composite electrolyte facilitated rapid refilling of surface oxygen vacancies and activation of molecular oxygen. Such rapid refilling of oxygen vacancies, in turn, accelerated lattice diffusion of oxygen ions [33,34]. The lattice oxygen vacancies (V_Ö_^″^) were along the (111) plane in FSZ, while PSZ had such vacancies (V_Ö_^″^) along the (211) and (110) planes [33]. For PSZ, the ($\bar{1}$01^M^) plane attributed to ZrO_2_ possessed 3-fold more V_Ö_^″^, and the ($\bar{1}$11^M^) plane possessed 2-fold more V_Ö_^″^, in comparison to the (211) and (110) planes. Figure 3a shows the absence of the ($\bar{1}$01^M^) and presence of the ($\bar{1}$11^M^) plane in the FSZ composite electrolyte signifying the reduced proportion of V_Ö_^″^. Such a condition results in difficulty in removing oxygen from the surface, thereby making the FSZ composite electrolyte less reducible and more stable [35]. The XRD patterns of as-received ∝-Al_2_O_3_ (Figure 3d) corresponded to a wide range of Bragg’s angles (18° ≤ 2θ ≤ 89°) attributed to a hexagonal phase (reference JCPD’s file no. 71-1123) with space group R$\bar{3}$c [36,37,38]. The average crystallite size obtained for ∝-Al_2_O_3_ via the Scherrer equation was near 26 nm [36,39].

Figure 3e, f shows that the diffraction patterns collected for Au and LSM did not register the presence of amorphous phases of gold oxide. Figure 3e shows a typical Au powder pattern with a preferred orientation along the (111) plane. The average crystallite size obtained for the Au particles was approximately 5 nm [40,41]. The LSM diffraction peaks indicated a perovskite structure with high crystallinity and a mixture of cubic and tetragonal phases [42]. The peak around 38.3° presented in both LSM and LSM-Au samples indicated the (202) plane (Sr doping level x ~0.22) that was consistent with a Sr doping level where x ≤ 0.3 in both cases [43]. As observed in Figure 3g, some weak features were detected at 21.2°, 28.2°, 44.4°, and 66.7° with 10 wt% loadings of Au powder, indicating the Au particles were well-dispersed over the LSM-Au electrode surface [42]. No signature for resistive La_2_Zr_2_O_7_ and SrZrO_3_ phases were observed in LaSrMnO_3_-Au/FSZ-PSZ-Al_2_O_3_ samples [41,44]. Such bimodal diffraction peaks in LSM-Au show less-improved crystalline symmetry, leading to the localization of the majority of the electrons, resulting in a hopping conduction mechanism at low-frequency regions [43]. The XRD data revealed the crystal phases and allied defects associated with the different crystal planes in the FSZ composite sensor material that impacted the overall oxygen vacancies and charge conduction mechanism, which affected the NO_x_ sensing. Further details on structural analysis can be found in Appendix A.

#### 3.1.3. Estimating the Chemical State of Structural Elements via XPS

The XPS in Figure 4a that contained the zirconia spectra with doublets Zr3d_5/2_ band (181.2 eV) and Zr3d_3/2_ band (183.5 eV) indicated the zirconia primarily consisted of monoclinic phases, as similarly observed in the XRD of PSZ (Figure 3b) [45]. Another doublet, Zr3d_5/2_ band (182.0 eV), and Zr3d_3/2_ band (184.5 eV) from the Zr-O bond was likely due to the FSZ composite electrolyte possessing tetragonal/cubic phases. Both doublets indicated that the zirconia present in the electrolyte had a Zr^4+^ state [45,46]. Figure 4a also shows Y3d_5/2_ (156.6 eV) and Y3d_3/2_ (158.7 eV), which corresponded to the tetragonal phase of the FSZ composite electrolyte [45,47]. The Y3d_5/2_ (157.2 eV) and Y3d_3/2_ (159.3 eV) indicated that the Y_2_O_3_ in the FSZ composite electrolyte primarily had cubic phases with Y^3+^ state cations within the yttria [45,47]. Figure 4b shows the Al2s peak, corresponding to the ∝-Al_2_O_3_ present in the FSZ composite electrolyte [48]. Figure 4c shows Sr3d_5/2_ (132.5 eV) and Sr3d_3/2_ (134.3 eV) peaks ascribed to the strontium (Sr^2+^) ions in the LSM lattice. Additionally, Sr3d_5/2_ (133.0 eV) and Sr3d_3/2_(134.8 eV) peaks were both associated with segregated SrO species on the LSM surface [42,49]. Figure 4d shows the primary lanthanum spectra of the LSM-Au sensing electrode, where doublets La3d_5/2_ (833.9 eV) and La3d_3/2_ (850.8 eV) lines occurred due to spin–orbit interaction and the electron transfer from oxygen ligands to the empty level La4f [49,50]. The double peaks for spin–orbital interaction reflected the La3d state configuration, which has an “M” mainline (ascribed to 3d^9^4f^0^L, L ascribed to oxygen ligand) and a satellite line S (ascribed to a charge transfer 3d^9^4f^1^L, where L represents the hole in a ligand site) at higher binding energies (see Figure 4d) [49,50]. The first splitting had a separation of ~16.9 eV attributed to LSM at the outer surface of the LSM-Au electrode. Another doublet La3d_5/2_ (835.3 eV) and La3d_3/2_ (852.0 eV) had a splitting separation of ~16.7eV, and the difference between the two La3d_5/2_ peaks was 1.4 eV for La_2_ O_3_, and that was attributed to LSM at the inner LSM-Au electrode surface [49,50]. Such spectral signatures were similar in detecting La at the LSM situated near the yttria-stabilized zirconia surface [51,52]. The additional doublet La3d_5/2_ (838.4 eV) and La3d_3/2_ (855.2 eV) peak lines were ascribed to shake up satellite features. The doublet of the La3d state separated by ~16.8 eV originated from the LSM-Au outer surface [50,51,52]. The La3d_5/2_ peaks had a difference (838.4−833.9 ~4.5 eV) ascribed to the La^3+^ state [52]. Another La3d satellite peak (848.0 eV) developed from a strong mixing of 3d^9^4f^2^ and 3d^9^4f^3^ states at the LSM-Au surface [50]. Figure 4e shows Mn2p_3/2_ (641.5 eV) and Mn2p_1/2_ (653.2 eV) were from the surface LaMn^3+^O_3_ species of the LSM-Au electrode. The Mn2p_3/2_ (644.1 eV) and Mn2p_1/2_ (655.3 eV) were ascribed to SrMn^4+^O_3_ species from the LSM-Au surface. The 2p_3/2_ and 2p_1/2_ spectral lines for Mn^3+^ species were separated by 11.7 eV, whereas the separation recorded for Mn^4+^ species was 11.2 eV [42,50,53]. The Sr^2+^ doping in lanthanum manganite affected the charge equilibrium by reducing the Mn^4+^ species to Mn^3+^ species in both LSM and LSM-Au materials. Such activity increased the overall relative at% of Mn^3+^ species to keep both materials electrically neutral (for details, see in Appendix A Appendix A) [42]. Figure 4f illustrates the spectral line Au4f_7/2_ (84.31 eV) with a difference of 3.67 eV with Au4f_5/2_ (87.98 eV) due to spin–orbit splitting attributed to reduced bulk metallic Au from the LSM-Au surface. Such splitting indicated the presence of some nanosized Au particles in the LSM-Au matrix [54]. The higher spin component Mn3s_1_ (82.44 eV) and lower spin component Mn3s_2_ (90.65 eV) with a splitting ~ 8.21 eV were much higher than the reported value. Such a scenario indicated relatively more Mn^4+^ oxygen complexes than Mn^3+^ complexes within the LSM-Au lattice structure [53]. The overabundance of Mn^4+^ species formed structural heterogeneity and clusters. Such heterogeneity resulted in over stoichiometric oxygen, suggesting interstitial oxygen within the cubic perovskite structure of LSM-Au, and thereby preserving sublattice oxygen and new oxygen structures [42]. The collective evidence from the XPS of the FSZ composite sensor revealed that the presence of ionic species associated with the materials provided more conducive pathways for electronic and ionic conduction, thereby facilitating interfacial electrochemical reactions during sensing at the FSZ composite NO_x_ sensor. Further details of XPS analysis are provided in Appendix A.

### 3.2. Working Principle for FSZ Composite NO_x_ Sensor

Conventional NO_x_ sensors typically incorporate a dense electrolyte and porous sensing electrode. However, enhanced sensing behavior has been observed at NO_x_ sensors composed of a porous electrolyte accompanied by a dense sensing electrode. The porous electrolyte enables gas diffusion and oxygen ion transport, while avoiding heterogeneous catalysis reactions observed at porous sensing electrodes. Heterogeneous catalysis can alter the concentration of the analyte gas and consequently reduce the accuracy of the NO_x_ response. The operating strategy and microstructural and compositional features are known to influence the working principles of solid-state gas sensors [32,55,56,57]. In the following paragraphs, the roles of the FSZ porous composite electrolyte and dense composite sensing electrode are discussed relative to impedimetric NO_x_ sensing. Oxygen ion transport, oxygen partial-pressure dependence, and charge-transfer mechanisms, which are understood to govern the materials composing the composite sensor, were used to describe the working principle of the FSZ composite NO_x_ sensor.

#### 3.2.1. Oxygen Ion Transport through the Porous FSZ Composite Electrolyte

The porous microstructure of the FSZ composite electrolyte enables two types of transport pathways. One pathway provided oxygen ion transport through the bulk material, and another allowed gas diffusion through the pores. Oxygen ion movement is understood to occur via the vacancy diffusion mechanism in the electrolyte. Here, the Y_Zr_^′^ impurity has a relative charge of −1, and the V_Ö_^″^ contains a relative charge of +2. The FSZ composite electrolyte under the electroneutrality condition results in (Y_Zr_^’^) ^≫^(e^−^). The electroneutrality condition is vital for retaining a constant oxygen vacancy concentration under reducing gas conditions. In the FSZ composite electrolyte, FSZ regions with high yttria doping could have trapped V_Ö_^″^ in Y_Zr_^’^ impurities (see Figure 5a), due to Coulombic interactions, thereby forming a quasi-stable defect pair as follows [32]:(1)$$Y_{\mathrm{Zr}}^{\prime }+V_{\ddot{o}}^{\prime \prime }↔{\left(Y_{\mathrm{Zr}}^{\prime }+V_{\ddot{o}}^{\prime \prime }\right)}^{\prime }\;$$

Such a defect pair may influence the free vacancy concentration and, thus, affect the ionic conductivity of the FSZ composite electrolyte [32]. As the ionic conductivity of FSZ was about 50% higher than that of PSZ [24], oxygen ion transport would not be as rapid via the PSZ particles within the FSZ composite electrolyte. Therefore, limited oxygen ion transport at PSZ particles, along with the presence of Al_2_O_3_ particles, may restrict electrochemical reactions with interfering gases in the composite electrolyte. A similar occurrence was reported for porous PSZ-based NO_x_ sensors where the presence of PSZ appeared to limit H_2_O cross-sensitivity in related studies [24]. Additional details regarding defects influencing ion transport through the FSZ composite electrolyte is provided in Appendix A Section S4.

#### 3.2.2. Charge Transport at LSM-Au Electrode and NO_x_ Sensing Mechanism

The electronic bandgap, the electron density of states at the surface, and the oxygen surface exchange properties of the LSM surface depend on strain, grain–grain boundary orientations, and extended defects of the sensor surface [25,26]. The strontium (Sr) doping introduced holes in the LSM lattice that helped in oxidizing NO. Doping of bivalent Sr^2+^ ions into the A-site for La^3+^ in the crystal lattice of lanthanum manganite resulted in the formation of equivalent amounts of Mn^4+^ ions or holes (see Figure 5b) [58]. The hole-hopping mechanism is usually prevalent in thermally activated charge transport of LSM at elevated temperatures >500 °C. Moreover, Sr^2+^ ion doping elevates the conduction pathway via oxygen species (Mn^3+^ ⇆ O^2#x2212;^ ⇆ Mn^4+^) near the triple-phase-boundary (TPB) region of the surface reaction site (see Figure 5b) [26,58]. The electron transfer occurred from a Mn^3+^ ion across intervening O^2-^ ion to the adjacent Mn^4+^ ion, or between two adjacent Mn^3+^ ions. Oxygen non-stoichiometry and electrical conductivity of the LSM-Au electrode agree with the random-defect model, as the oxygen reduction reaction resulted in the maximum current flow through the TPB region at the LSM surface. The hole formation enhanced electrical conductivity via a double-exchange phenomenon facilitating the Mn^4+^ ions to migrate to the TPB region of LSM to oxidize the NO to produce NO_2_ [58]. During the consumption of NO along with oxygen species O^2#x2212;^, the electron trapped by O^2#x2212;^ is transferred to the TPB region at the LSM-Au surface, thus reducing the number of hole carriers via electron–hole recombination [58,59]. One-third of the active holes are concentrated within the Mn sublattice, while the remaining two-third of holes are in the oxygen sublattice [26]. The oxygen sublattice serves as a dynamic source of holes during electron–hole recombination. Such recombination occurs during lattice oxygen vacancy formation that is prevalent in perovskites [25,26,27]. The Au particles integrated within the LSM electrode act as an electron sink, separating electrons and holes and, thus, reducing the charge carrier recombination rate within the LSM-Au matrix [42,58]. Another benefit of Au as a composite sensor material is that it is chemically inert, and, in contrast to Ag or Pt, Au does not catalyze oxygen reduction reactions [59].

#### 3.2.3. Operating Mechanism of Solid-State Electrochemical FSZ Composite NO_x_ Sensor

During impedimetric sensor operation, NO_x,_ and O_2_ gases adsorb onto the electrolyte surface and undergo dissociation, as well as diffuse through the porous electrolyte of the FSZ composite NO_x_ sensor to the active sites at the TPB where interfacial electrochemical reactions take place. At a temperature >500 °C, electrochemical reactions and oxygen reduction reactions proceed more readily, thereby aiding O^2#x2212;^ ion transport through the electrolyte bulk; and O^2#x2212;^ ion generation occurs via charge transfer reactions at TPB.

In studies by Rheaume et al., the NO_x_ sensing mechanism was reported to be governed by competitive molecular dissociative adsorption of both NO_x_ and O_2_ at the active sites at TPB, causing a suppressed impedance arc [60]. Such competitive adsorption results in the charge transfer process at the TPB that is mapped at lower frequencies. The lower-frequency-arc width associated with charge-transfer resistance from redox reaction consists of adsorption, dissociation, and interfacial electrochemical reactions due to NO_x_ and O_2_ sensitivity at the TPB active sites. Upon exposure to NO_x_, the charge-transfer resistance reduces with an inward shift in the lower-frequency arc. Such a shift is primarily caused by reversible adsorption of NO_x_ species at the TPB active sites. The inward shift can also occur due to increased O_2_ diffusion via saturating the TPB active sites. Saturation may result in decreasing magnitude in the phase angle during NO_x_ sensing, thus resulting in less sensitivity. The rate-determining step of NO_x_ sensing also depends on other oxygen-species-related reactions at TPB, such as the electron transfer occurring in the LSM-Au surface between two adjacent Mn^+^ ions and trapped O^2#x2212;^ ion, which is transported through the FSZ composite electrolyte.

##### Electrochemical Behavior LSM-Au/FSZ Composite NO_x_ Sensor

The electrochemical responses of the FSZ composite sensors were characterized by using the impedimetric method. For comparison purposes, the non-composite FSZ sensors were also evaluated. The sensors were operated with 10.5% O_2_, both with and without 100 ppm NO and NO_2_ present, where N_2_ served as the background gas. Since the electrochemical response of the sensors exposed to NO and NO_2_ was similar, the impedance-based results presented for the sensor focused on operation with NO.

For each sensor type, two distinct frequency regimes were apparent, as shown by the arcs at high (i.e., >10 kHz) and low (i.e., <1000 Hz) frequencies in the impedance data of Figure 6. The FSZ composite sensor resulted in a more significant impedance response than the FSZ sensor response. It was also observed that adding 100 ppm NO to the gas stream caused the magnitude of the low-frequency impedance arc to decrease for both the FSZ composite and FSZ sensors.

The equivalent circuit model is included in Figure 6. The high-frequency data were described by R_HF_∥Q_HF_, where R_HF,_ the electrolyte resistance, and Q_HF_ represent non-ideal capacitive effect. The low-frequency data were denoted by R_LF_∥(C_LF_ + W), where the resistance, R_LF_, was modeled in parallel with a capacitor, C_LF_, in series with a Warburg (W) diffusion element. The Q_HF_ arises from the inhomogeneous nature of the porous electrolyte that can be traced in respective bode plots (see details in Appendix A Section S5) [10,60]. In addition, the irregularly shaped coarse particles evident in SEM images of the FSZ composite (see Figure 2b,c) also contributed to the inhomogeneous microstructure of the FSZ composite sensors.

The impedance describing the high-frequency data, Z_HF_, in the equivalent circuit model of FSZ composite electrolyte is described by the following complex equation [22,24,32,61]:(2)$$Z_{\mathrm{HF}}=\frac{R_{\mathrm{HF}}}{1+R_{\mathrm{HF}}Q_{\mathrm{HF}}{\left(j\omega \right)}^{n}}$$
where j = √−1, ω = 2πf, and f is the sensor operating frequency. The non-ideality factor, n, determines the fit of Q_HF_ and represents the depressed semicircle nature of the arc in the porous electrolyte. At high frequencies where 1 ≪ R_HF_Q_HF_(jω)^n^, oxygen ions usually cannot penetrate through the porous pathways of the electrolyte [61,62]. The impedance ZQ_HF_ contributed by Q_HF_ is described below:(3)$$Z_{Q_{\mathrm{HF}}}=\frac{1}{Q_{\mathrm{HF}}{\left(j\omega \right)}^{n}}\;$$

For the high-frequency regime measurement, the Q_HF_ value was ~3.1 nF.s^n−1^ for the FSZ composite sensors operating with 0 and 100 ppm NO_x_, as the rate-limiting mechanism was independent of the gas concentration (see Table 1). The FSZ sensors had comparably a substantially lower Q_HF_ value of ~0.21 nF.s^n−1^ on account of the fine particles and pores composing the FSZ electrolyte microstructure. The n value of ~0.75 for the FSZ composite sensors indicated the dispersive nature of the capacitive dielectric coupling in the FSZ composite electrolyte. It also indicated a decrease in charge conduction at a higher frequency region, due to the skin effect, where charged ions are more scattered at the surface than the bulk of the FSZ composite electrolyte [43]. In contrast, the FSZ electrolyte had an n value of ~0.84, indicating a stronger hopping charge conduction mechanism [43].

The capacitance at high frequency for the FSZ composite sensors can be calculated from circuit parameters R_HF_ and Q_HF_ via the following formula [32]:(4)$$C_{\mathrm{HF}}=R_{\mathrm{HF}}^{1-n}\times Q_{\mathrm{HF}}^{\frac{1}{n}}\;$$

The C_HF_ values for both the FSZ and FSZ composite electrolytes were in the Femto farad range, signifying charges diffused through both electrolytes. The FSZ composite electrolyte resulted in ~3-fold higher C_HF_ values (see Table 1) that indicated the inclusion of PSZ and ∝-Al_2_O_3_ caused a comparably higher time constant for the ion-transfer process, which signifies the faster ion-transfer rate in the FSZ composite sensor [32].

The impedance data at low frequencies described electrode and electrode/electrolyte interfacial reactions occurring within the sensors. The formation of surface oxygen vacancies, O_o_^x^, at the LSM-Au electrode relied on adsorbed oxygen release and Mn^4+^ ions reduction. The O_o_^x^ promoted dissociative adsorption of oxygen and increased the oxygen rate at the LSM-Au surface, resulting in enhanced migration of surface oxygen species, thus leading to charge transfer at the triple phase boundary [10,32]. The charge-transfer resistance (R_LF_) value was in the range of ~34.7 kΩ (at 0 ppm NO) and to ~28.5 kΩ (at 100 ppm NO), showing an increase in rate-transfer kinetics for the LSM-Au electrode and the FSZ composite electrolyte interface. Since R_LF_ was more extensive in magnitude at 0 ppm, in comparison to 100 ppm, the dense LSM-Au surface likely contributed toward the higher impedance observed in bode plots (Appendix A in Appendix A Section S5). The interfacial double-layer capacitance, C_LF_, arose primarily from grains at the dense surface LSM-Au at the interface. The C_LF_ value varied from 0.62 µF (at 0 ppm NO) to 0.40 µF (at 100 ppm NO). The fitting of the interface with capacitance indicated a hopping type of conduction mechanism with localized electrons at the interface [43].

The low-frequency impedance, Z_LF_, was described according to the following equation:(5)$$Z_{\mathrm{LF}}=\frac{R_{\mathrm{LF}}}{1+{j\omega R}_{\mathrm{LF}}C_{\mathrm{LF}}}\;$$

The fitting data in Table 1 show that the R_LF_ values are on the order of kΩ, and C_LF_ values are on the order of microfarads. So, for all frequency ranges, the denominator of Equation (5) satisfies 1 ≫ ωR_LF_C_LF_ condition, which simplified the relation to Z_LF_ ≈ R_LF_ [61,62]. Such a scenario arises from the less electroactive region of the dense LSM-Au surface, where the impedance was not dependent on the frequency of the AC signal. The input current in such a region was equal to the output current, signifying the resistive region. The R_LF_ was also associated with the exchange current, I_0_, under equilibrium conditions, according to the following equation [63]:(6)$$R_{\mathrm{LF}}=\frac{\mathrm{RT}}{{\mathrm{nFI}}_{0}}\;$$
where R is the universal gas constant, F is Faraday’s constant, T is the temperature in Kelvin, and n represents the number of electrons transferred. The exchange current varied proportionally with the heterogeneous electron transfer rate constant, k_0_, as shown in the following equation [63]:(7)$$I_{0}=nFAk_{0}{\left(C\right)}_{\mathrm{bulk}}\;$$

For sensors in the present study, the number of electrons transferred per molecule of NO_x_ was n, the active area of LSM-Au under NO_x_ sensing was A, and the analyte (NO_x_) concentration was (C)_bulk_. During NO_x_ exposure, both the FSZ composite and FSZ sensors response indicated low R_LF_ values, in comparison to sensor operation without NO_x_, which signified enhancement in the electron transfer rate, k_0_, at the electrode-electrolyte interface [10,63]. Moreover, the charge transfer kinetics were more rapid for FSZ composite sensors, as the corresponding R_LF_ value was about 17 kΩ lower than R_LF_ determined for FSZ sensors, as shown in Table 1, thereby indicating that k_0_ was approximately 2.4 times faster for the FSZ composite sensors at the interface.

The Warburg circuit element, W, included in the circuit model shown in Figure 6, was associated with a Warburg impedance, Z_*w*_, resulting from the mass transfer at the electrode/electrolyte interface [10,61,62]. The concentration of NO_x_ influenced the diffusion at lower frequencies. The following equation describes Z_*w*_ as follows:(8)$$Z_{w}=A_{w}\left(1-j\right)\omega^{-\frac{1}{2}}\;$$
where A_w_ is the Warburg coefficient based on a semi-infinite diffusion model, j is the unit imaginary number, and ω is the angular frequency [10,62]. At the lower frequency range, A_w_ for 0 ppm NO was ∼240 kΩ∙s^−0.5^ and ∼118 kΩ∙s^−0.5^, for the FSZ composite and FSZ sensors, respectively. Minimal variation in A_w_ was observed upon NO_x_ diffusion for 100 ppm NO indicated high-oxygen surface-exchange properties at the LSM-Au sensing electrode [43]. The lower A_w_ values obtained for the FSZ sensors indicated that mass transport was more rapid at the electrode/electrolyte interface than the FSZ composite sensors. Such rapid mass transport was most likely due to the faster diffusion of oxygen ions through the FSZ electrolyte to the electrode/electrolyte interface, where NO_x_ sensing reactions took place. Since the XRD and XPS (details in Section 3.1.2 and Section 3.1.3) did not show any signatures of resistive phases, such as La_2_Zr_2_O_7_ or SrZrO_3_, it was assumed that the Warburg impedance and the charge transfer resistance had no contributions from such resistive phases [41,44]. As the LSM-Au electrode is not an oxygen ion conductor, the oxygen reactions were localized to the surface regions at the TPB [43].

#### 3.2.4. NO Sensitivity of FSZ Composite NO_x_ Sensor

We used angular phase response, θ, to evaluate NO sensitivity, as it can provide a more accurate and stable measure of the sensor response to the analyte gas, in comparison to other impedance components, such as the modulus, |Z| [7,10,56]. The change in the angular phase response that occurred when NO was added was used to determine the sensitivity to NO based on the following equation:(9)$${\mathrm{NO}}_{\mathrm{sensitivity}}=\frac{\Delta \theta }{\Delta \mathrm{NO}}\;\left(\frac{\mathrm{degrees}\;\left({}^{\circ}\right)}{\mathrm{ppm}}\right)\;$$
where Δθ = θO_2_ − θ_NO_, and the terms θO_2_ and θ_NO_ correspond to the angular phase response when the sensor was exposed to O_2_ + N_2_ and O_2_ + NO + N_2_, respectively. The angular phase response with and without 100 ppm NO was measured for sensors operating at temperatures ranging from 575 to 675 °C, as shown in Figure 7. The largest Δθ values were obtained by operating the sensors at a frequency of 20 Hz and a temperature of 575 °C. Thus, these conditions were chosen to assess the NO sensitivity of the NO_x_ sensors.

The angular phase response for both FSZ composite and FSZ sensors is shown in Figure 8 for dry and humidified gas conditions with NO concentrations ranging from 0 to 100 ppm for sensor operation at 20 Hz and 575 °C. Both types of sensors followed linear trends and had a high and low sensitivity range. The related R^2^ values are presented in Appendix A Section S6. The FSZ composite sensors demonstrated the most significant sensing response to NO, which was about 0.071°/ppm for NO concentrations ≤25 ppm under dry gas conditions. As the NO concentration increased, the sensing response decreased to approximately 0.040°/ppm. The decrease in the sensing response indicated gas-saturation effects. Under similar conditions, the FSZ sensors demonstrated a more comprehensive high sensitivity range that extended up to 50 ppm NO, but such high sensitivity was accompanied by a lower NO sensing response of 0.051°/ppm that decreased to about 0.036°/ppm for the higher NO concentration range.

Adding water to the gas stream caused the NO sensitivity to increase for both sensor types (see Figure 8). The water cross-sensitivity of the FSZ composite sensors diminished with increasing operating temperatures beyond 575°C (details in Appendix A Section S7). However, NO sensitivity also decreased as the sensor operating temperature increased (see Figure 7). In other studies, the authors explored the influence of particle size on water cross-sensitivity [24]. The study was carried out by comparing the impedance response of porous electrolyte NO_x_ sensors composed of fine PSZ particles to coarse PSZ particles for operation in dry and humidified gas conditions [22,24]. The data observed indicated that these sensors behaved similarly in the presence of water [24]. The water cross-sensitivity study suggests that the size of the PSZ particles composing the porous electrolyte did not influence water cross-sensitivity at these sensors. Water-adsorption experiments conducted in other studies have reported that molecular water can strongly adsorb onto the Y_2_O_3_ phases and hydroxyl groups form due to surface reactions [64,65]. In addition, computational studies have indicated that molecular water dissociation is a mechanism for hydroxyl species formation at Y_2_O_3_-ZrO_2_ surfaces and interfacial reactions involving the oxide and hydroxyl groups can enhance oxygen ion conductivity [66]. The enhanced oxygen ion conductivity for both sensors in the present study promoted the water cross-sensitivity. The humidified gas environment impacted FSZ composite sensors more than FSZ (see Figure 8). Reducing water cross-sensitivity may be possible by altering the composition of Au particles in the LSM-Au sensing electrode, as Au electrodes have been found to limit water cross-sensitivity [15].

#### 3.2.5. The O_2_ Influence on NO Sensitivity of The Composite NO_x_ Sensor

Although oxygen reactions resulting in oxygen ions contribute to electrochemical NO_x_ reactions, competition between O_2_ and NO for active sites along the TPB can interfere with the NO sensitivity. Such an interference can limit sensor accuracy. Thus, the influence of oxygen on NO sensitivity was evaluated by operating the sensors under various oxygen environments. Figure 9 shows a graph of the NO sensitivity for the oxygen concentration present in the gas stream for both types of sensors. Compared to the FSZ sensors, the FSZ composite sensors were far less sensitive to oxygen concentration changes over a range of 5–18 vol%. Such results indicate that the oxygen cross-sensitivity was comparatively less significant for the FSZ composite sensors. (In Appendix A Section S8, the rate-limiting mechanisms due to the low oxygen dependence are discussed.)

The FSZ sensors had a higher density of TPB sites relative to the FSZ composite sensors, where the Al_2_O_3_ addition blocked electrochemical reaction sites between the electrolyte and LSM-Au sensing electrode [23]. Thus, the FSZ sensors provided more opportunities for O_2_ reactions to proceed at the electrode/electrolyte interface, promoting oxygen dependence. In other NO_x_ sensor studies, mixed conductivity (i.e., ionic, and electronic conducting) sensing electrodes were found to have greater cross-sensitivity to oxygen relative to gas sensors with purely electronic sensing electrodes [10]. The mixed conductivity of the electrode enlarged the TPB region, providing a greater density of sites where oxygen reactions proceeded readily. The conductivity data, along with the observations of the present study, suggested that NO_x_ sensors with a sufficiently high density of active sites for oxygen reactions could be more prone to O_2_ cross-sensitivity.

#### 3.2.6. The Influence of CH_4_, CO_2_, and CO on NO sensitivity

In addition to O_2_ cross-sensitivity, interference from other exhaust gases can limit the NO_x_ sensor response accuracy. The NO_x_ sensitivity was evaluated under operation with CH_4_, CO, and CO_2_ gases, as shown in Figure 10, where the response to these gases was compared to the sensitivity measured for NO and NO_2_ gases. The Δθ measurements shown in Figure 10 are based on the gas response with and without 100 ppm of the analyte gas (NO) present and air as the background gas. Both sensor types showed negligible responses to CO and CO_2_, although the change of the angular phase response was in the opposite direction of the NO and NO_2_ response. The FSZ composite sensors were also less prone to CH_4_ cross-sensitivity in comparison to the FSZ sensors. The response to NO and NO_2_ was approximately 3 times greater for FSZ composite sensors than FSZ sensors.

More yttria in the FSZ sensor decreased the surface lattice oxygen ions, essentially removing surface oxygen ions and promoting partial catalytic methane oxidation [33]. Moreover, CO_2_ reforming of methane to carbon monoxide (CH_4_ + CO_2_ → 2CO + H_2_O) and the reverse water gas shift reaction (CO_2_ + H_2_ → CO + H_2_O) could simultaneously occur [42]. The presence of Al_2_O_3_ in the FSZ composite electrolyte limited the active sites along with the electrode/electrolyte interface, which subsequently reduced the activity for CO and CO_2_ oxidation. Difficulty in removing lattice oxygen from the surface of the FSZ composite electrolyte caused it to be less reducible and more stable, and, thus, not conducive for CO and CO_2_ partial oxidation at the sensor surface [35].

Such susceptibility toward oxidation of CO and CO_2_ indicates that operating the FSZ sensor in the air can substantially hinder its NO_x_ sensing capability. Such an observation agreed with the findings in Figure 9, which indicated that the composite sensors were less susceptible to variations in the oxygen concentration. Moreover, LSM-based sensing electrodes are known to promote high NO_x_ sensitivity and selectivity in high-oxygen environments [67]. Such characteristics and the low oxygen dependence of the FSZ composite sensors most likely contribute to the sensor selectivity. The dominant sensing behavior of the FSZ composite sensors relative to the FSZ sensors demonstrated that the composite sensors could provide a more accurate NO_x_ sensing response.

#### 3.2.7. Composite NO_x_ Sensor Response/Recovery Rates

To further explore the accuracy of the FSZ composite sensors, time-based measurements of the NO sensing response and recovery rates were collected. Figure 11 shows the change in the angular phase response for an FSZ composite sensor operating at 20 Hz at T = 575 °C, while the NO concentration changed from 0 to 100 ppm. The response and recovery time, τ_90_, was determined to be within 34 to 36 s for each of the concentrations measured. The LSM-Au sensing electrode may limit the FSZ composite sensor in response and recovery times. In other studies, NO_x_ sensor studies with LSM and LSM-YSZ sensing electrodes were found to have average response times of 15 to 17 s, respectively [10]. However, these sensing electrodes demonstrate substantially lower NO_x_ sensing capabilities relative to the LSM-Au sensing electrodes. It may be possible to improve the response/recovery time of the LSM-Au sensing electrode without sacrificing NO_x_ sensitivity by modifying the concentration ratio of LSM to Au composing the electrode. Another approach could be to substitute the solid LSM particles with hollow/porous LSM microparticles as the inherent pore size distribution is thought to enhance the response and recovery rates of LSM-based sensors [68]. Further observation of the data in Figure 11 indicated that the FSZ composite sensor response resolution was achieved for NO concentrations as low as 2 ppm. Such an outcome was likely related to the low O_2_ cross-sensitivity of the FSZ composite sensor, and the high NO_x_ sensitivity promoted by the LSM-Au sensing electrode.

## 4. Conclusions

The microstructures of both the NO_x_ sensor electrolyte and sensing electrode reveal that the defects and oxygen vacancies significantly affected TPB interfacial reactions that influence sensing behavior. The collective evidence from the XPS of FSZ composite electrolyte revealed the presence of Zr^4+^ and Y^3+^ species, which indicated the formation of defects that provided more extrinsic oxygen vacancies conducive for ionic conduction, as also supported by XRD. The XPS of the LSM-Au sample indicated that, along with La^3+^ and Sr^2+^ species, Mn^3+^ and Mn^4+^ species at the surface-provided electron-transfer pathways at TPB during NO_x_ sensing. The presence of Mn^4+^ species enhanced the double-exchange phenomenon, which facilitated the oxidation of NO at TPB during NO_x_ sensing. The working principle of the FSZ composite sensor involved the conductivity of the FSZ composite electrolyte, due to the defect concentration of ionic charge carriers, resulting in oxygen vacancies for charge compensation. Furthermore, during NO sense, the electrons trapped by O^2−^ were transferred to the TPB region at the LSM-Au surface, reducing the number of hole carriers recombining the electron–hole. The TPB in the FSZ sensors were composed of a high density of reaction sites that seemed to support both O_2_ and NO_x_ reactions, thereby enabling O_2_ cross-sensitivity. At the FSZ composite sensors, the TPB reaction was interrupted by Al_2_O_3_ particles present in the composite electrolyte which blocked active electrochemical reaction sites. Such a blocking of active sites significantly affects oxygen reactions, whereas NO_x_ reactions seem to proceed readily. The LSM-Au sensing electrode promoted NO sensitivity, while the FSZ composite electrolyte limited cross-sensitivity toward CO, CO_2_, and CH_4_ gases. The LSM-Au microstructure hindered the response and recovery rates. Modifications to the LSM microstructure or changing the ratio of LSM to Au for the electrode could potentially improve these rates. Such methods are under consideration for future studies. Investigations regarding the impact of H_2_O cross-sensitivity, which was evident for both sensor types via altering the amount of Au compositing the LSM-Au sensing electrode, are also under the scope of future study. Overall, the electrochemical behavior of the FSZ composite sensors indicated that the composite microstructure enabled a high NO_x_ sensing response and low cross-sensitivity to interferent gases, while also promoting NO detection capabilities down to 2 ppm.

## Figures and Tables

**Figure 1 materials-15-01165-f001:**
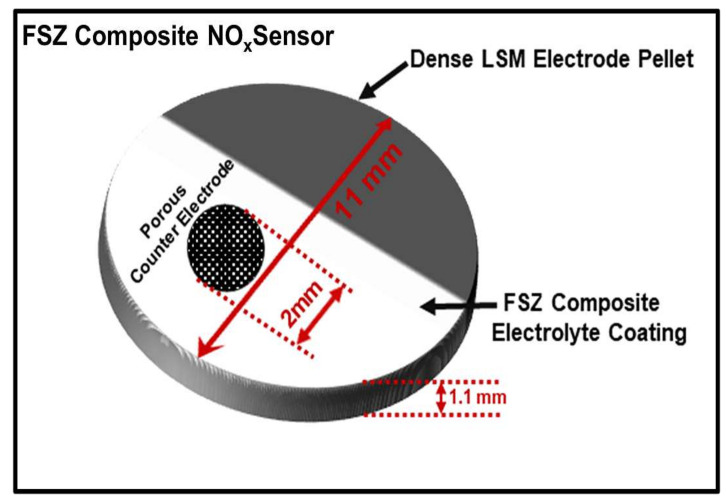
Schematic diagram of the FSZ composite NO_x_ sensor.

**Figure 2 materials-15-01165-f002:**
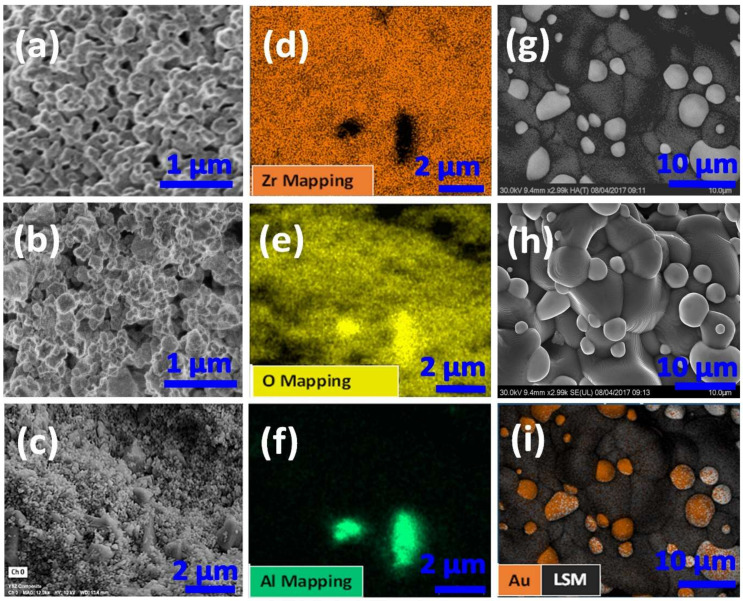
Typical SEM images of (**a**) FSZ porous electrolyte structure; (**b**) FSZ composite electrolyte porous structure; (**c**–**f**) FSZ composite electrolyte cross-section and corresponding elemental mapping via EDS for FSZ composite electrolyte tracing zirconia (Zr), oxygen (O), and aluminum (Al) at the composite electrolyte surface, respectively. (**g**–**i**) High-angle backscattered image of LSM-Au electrode, corresponding secondary SEM image of LSM-Au electrode, and elemental mapping via EDS of LSM-Au electrode cross-section.

**Figure 3 materials-15-01165-f003:**
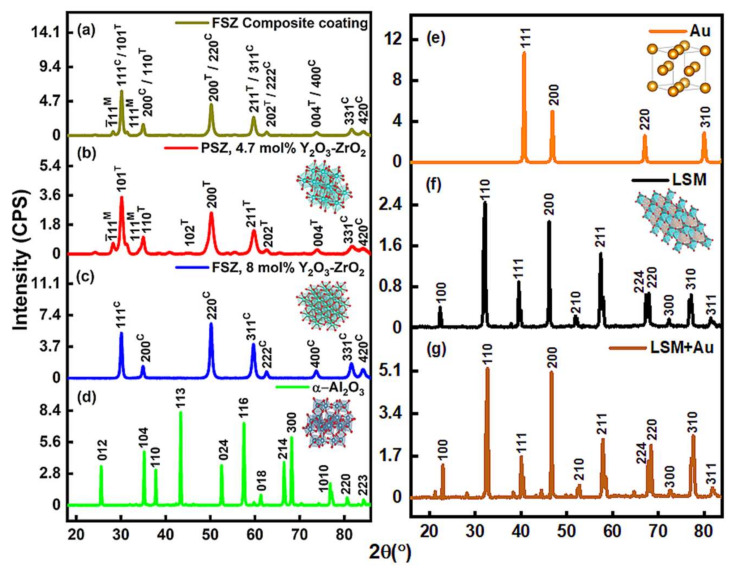
XRD diffraction of (**a**) FSZ composite coating, (**b**) PSZ, (**c**) FSZ, (**d**) Al_2_O_3_, (**e**) Au powder, (**f**) LSM, and (**g**) LSM-Au electrodes. The XRD diffraction in the FSZ composite electrolyte and the LSM-Au electrode estimates different crystal phases and related defects, influencing the ion transport mechanism during NO_x_ sensing. Inset arbitrary polyhedral crystal figures in (**b**–**f**) shows that the lattice structures of PSZ, FSZ, Al_2_O_3_, Au powder, and LSM, respectively, were drawn in VESTA version 3.4.8 (Amakubo, Tsukuba-shi, Japan; Copyright (©) 2006–2021, Koichi Momma and Fujio Izumi).

**Figure 4 materials-15-01165-f004:**
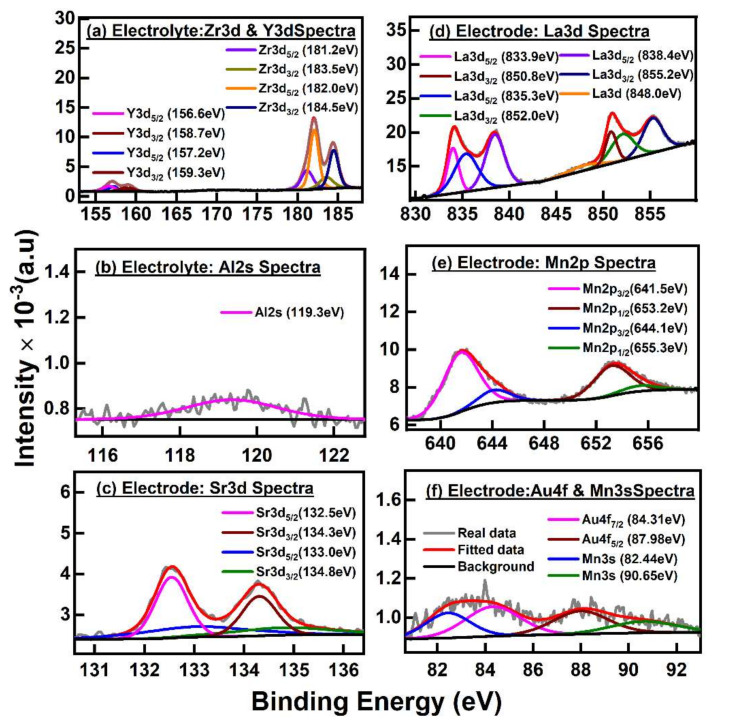
XPS spectra for FSZ composite electrolyte (**a**,**b**) and LSM-Au electrode (**c**–**f**) show the core elements’ chemical state for the composite NO_x_ sensor. The FSZ composite sensor (**a**) shows Y3d and Zr3d spectra indicating the presence of zirconium and yttrium, along with (**b**), signifies the presence of Al_2_O_3_ (alumina) within the electrolyte matrix. The LSM-Au hybrid electrode indicates the presence of (**c**) Sr3d for strontium, (**d**) La3d for lanthanum, (**e**) Mn2p for manganese, and (**f**) Au4f and Mn3s for gold and manganese in the hybrid electrode surface. Legends for individual spectra are indicated within respective figures. Each figure contains actual data (gray curve), fitted data (red curve), and background (black curve).

**Figure 5 materials-15-01165-f005:**
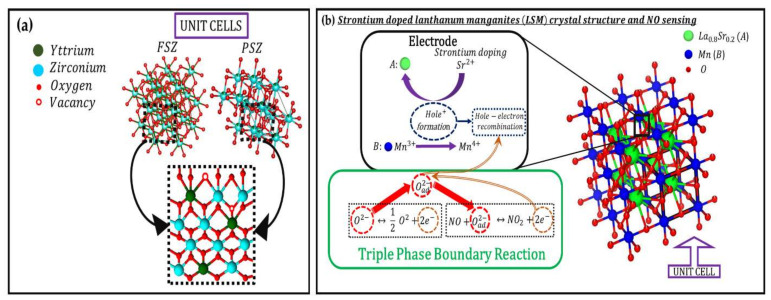
(**a**) 3D schematic ball and stick model of FSZ, and PSZ (2 layers, XYZ direction), illustrating the most likely defect model scenario in the FSZ composite electrolyte driving the ion transport mechanism during NO_x_ sensing. (**b**) Schematic diagram showing the effect of strontium doping in LSM lattice (ball-and-stick model) and subsequent charge-transfer mechanism at TPB initiated due to NO_x_ sensing. All relative crystal models in (**a**,**b**) are drawn in VESTA version 3.4.8.

**Figure 6 materials-15-01165-f006:**
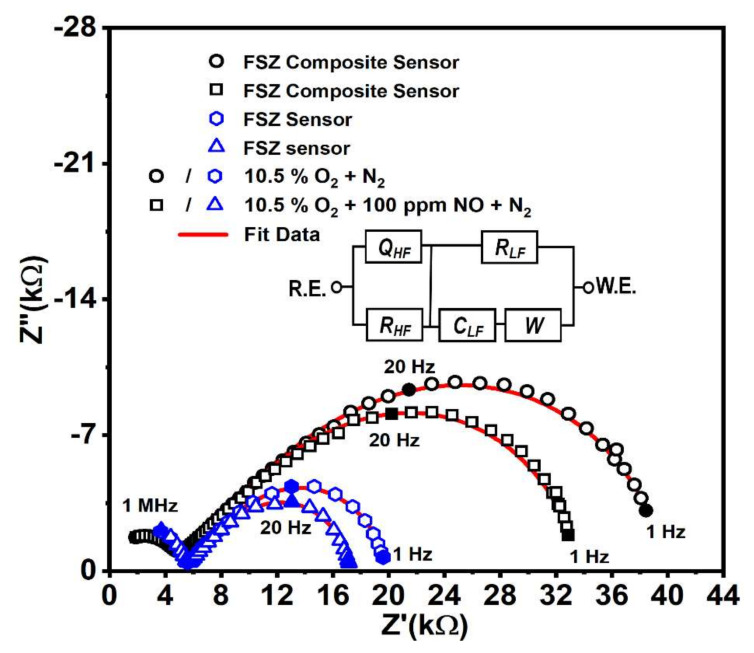
Impedance response of a NO_x_ sensor supported both FSZ electrolyte and FSZ composite electrolyte for operation at 575 °C with and without 100 ppm NO present. The equivalent circuit represents both types of NO_x_ sensors.

**Figure 7 materials-15-01165-f007:**
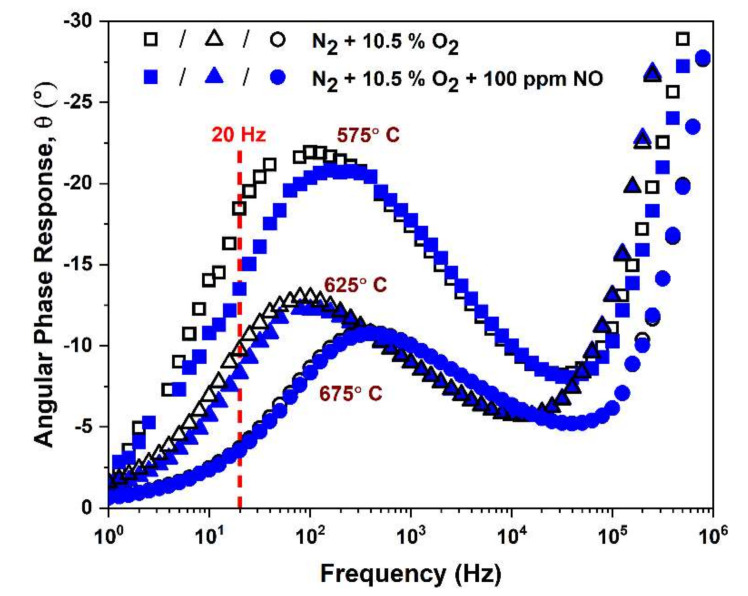
FSZ composite NO_x_ sensors angular phase responses at different operating temperatures with and without 100 ppm NO present. The figure shows that 575 °C is favorable because the low-frequency phase difference due to NO is significant.

**Figure 8 materials-15-01165-f008:**
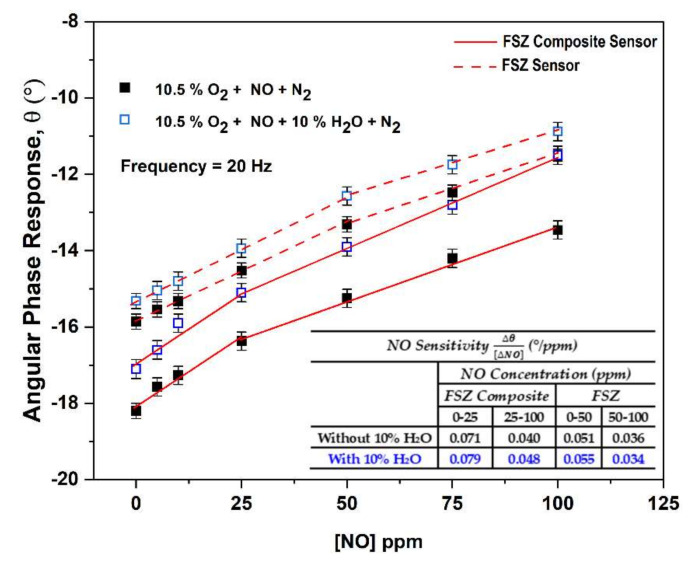
NO sensitivity of FSZ composite (hollow blue box) and FSZ electrolyte (black box) NO_x_ sensors at 575 °C for dry and humidified operating conditions. Inset table shows that FSZ composite NO_x_ sensors possess enhanced NO sensitivity compared to FSZ sensors in both dry and humidified conditions.

**Figure 9 materials-15-01165-f009:**
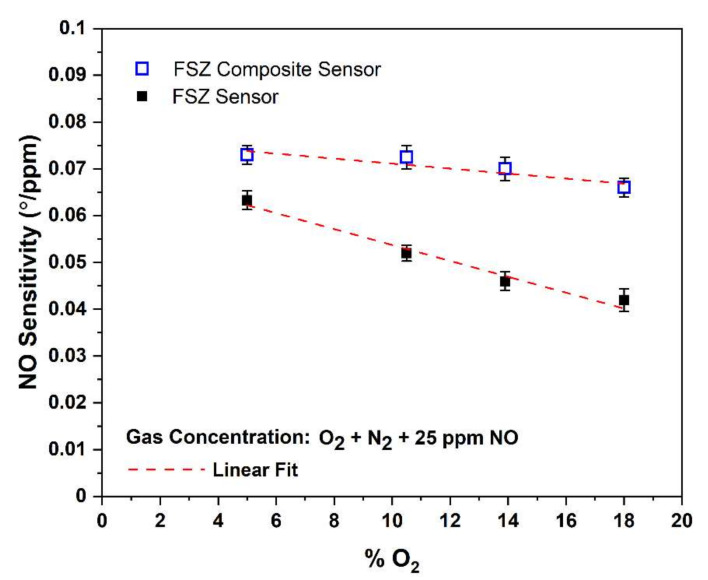
NO sensitivity dependence to the oxygen present in the gas stream for FSZ composite (hollow blue box) and FSZ (black box) NO_x_ sensors were operating at 20 Hz at 575 °C. The R^2^ values for FSZ sensor and FSZ composite sensor fittings are 0.97482 and 0.8729, respectively.

**Figure 10 materials-15-01165-f010:**
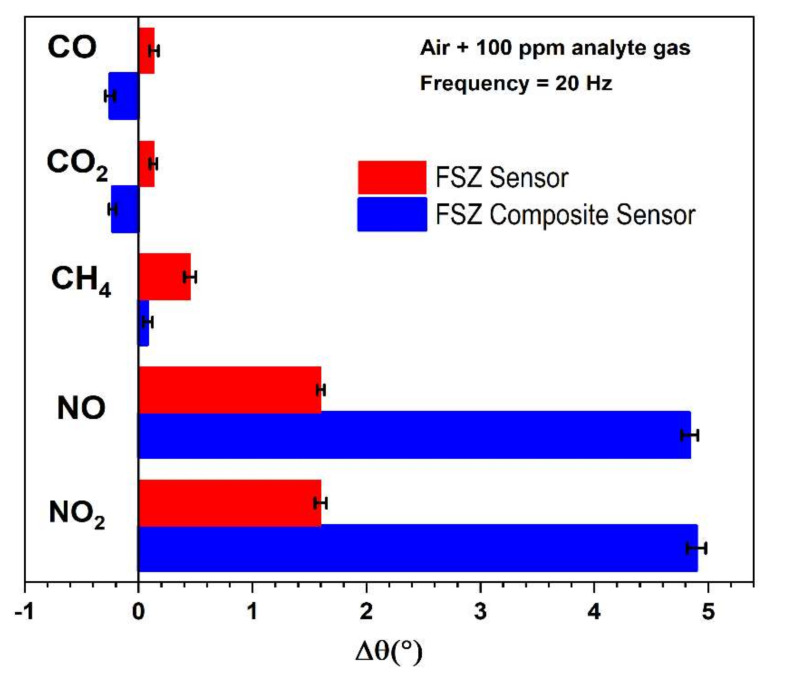
Sensing responses at 20 Hz frequency of FSZ (red bar) and FSZ composite (blue bar) sensors for operation with various interferent gases such as carbon monoxide (CO), carbon dioxide (CO_2_), methane (CH_4_), and nitrogen dioxide (NO_2_) at 575 °C.

**Figure 11 materials-15-01165-f011:**
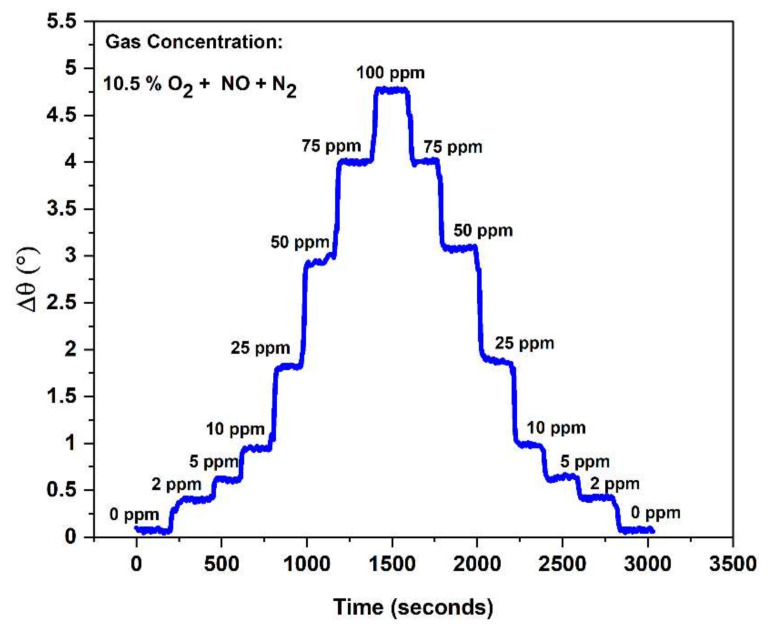
Time-based response of the FSZ composite NO_x_ sensor operating at 575 °C and 20 Hz shows high NO sensitivity down to 2 ppm.

**Table 1 materials-15-01165-t001:** Circuit fitting parameters for the Nyquist plot of the FSZ composite and FSZ sensor with and without NO_x_.

Circuit Parameters						
The FSZ Composite Electrolyte with the LSM-Au Electrode						
NO_x_	Electrolyte Impedance			Electrode-Electrolyte Interface		
	R_HF_ (kΩ)	Q_HF_ (nF.s^n−1^)	n	C_LF_ (μF)	R_LF_ (kΩ)	A_W_ (kΩ.s^−0.5^)
0 ppm	4.97 ± 0.75	3.17 ± 0.01	0.75 ± 0.5	0.62 ± 0.0	34.7 ± 0.5	240.6 ± 0.04
100 ppm	4.95 ± 0.8	3.17 ± 0.09	0.75 ± 0.5	0.40 ± 0.0	28.5 ± 0.9	239.5 ± 0.05
The FSZ electrolyte with the LSM-Au electrode						
NO_x_	R_HF_ (kΩ)	Q_HF_ (nF.s^n−1^)	n	C_LF_ (μF)	R_LF_ (kΩ)	A_W_ (kΩ.s^−0.5^)
0 ppm	5.43 ± 0.54	0.215 ± 0.00	0.84 ± 0.5	0.61 ± 0.0	14.3 ± 0.7	118.1 ± 0.04
100 ppm	5.46 ± 0.59	0.176 ± 0.00	0.85 ± 0.5	0.46 ± 0.0	11.7 ± 0.8	122.1 ± 0.06
The capacitance calculated for Electrolytes from Equation (4)						
NO_x_	FSZ composite electrolyte			FSZ electrolyte		
	C_HF_ (fF)			C_HF_ (fF)		
0 ppm	40.9			13.6		
100 ppm	42.2			14.3		

## Data Availability

Not applicable.

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
