# Peer review of "Investigation of an Impedimetric LaSrMnO3-Au/Y2O3-ZrO2-Al2O3 Composite NOx Sensor"

_materials, 2022, doi:10.3390/ma15031165_

Round 1
Reviewer 1 Report
The manuscript details the fabrication and thorough characterization of a new composite electrolyte used for impedimetric NOx sensing. The paper is very well written and I'd like to see it published after the following points are addressed:
- I would suggest using the term „impedimetric“ for sensors based on impedance measurements. It is a term which is used much more often, to the best of my knowledge.
- Please check Figure 6 position, it comes right after Figure 1.
- Line 551: You say the sensors „followed linear trends“, in two ranges. Please give R2 values to demonstrate this and discuss. Perhaps a different curve can be fit throughout the entire NO concentration range for the composite sensors?
- Line 620 and Figure 10: „The response to NO and NO2 was approximately 3 times greater for FSZ composite sensors than FSZ sensors“. This seems inconsistent with the sensor responses to 100 ppm NO at same frequency and temperature shown in Figure 8 - in Fig8 Δθ seems to be around 4-5 ° for both types of sensors. Please explain
- Please compare the performance of your sensor to some examples from the literature and/or comment whether performance (sensitivity, dynamic range, response time etc.) is suitable for the proposed application (diesel engine exhaust gas monitoring).
Author Response
We appreciate the valuable comments from the reviewer, which have helped us improve the clarity and quality of our manuscript. The revised manuscript considers the comments and suggestions from the reviewer, and the changes made in the revised manuscript are highlighted in blue color. Below, we provide a point-by-point response in the attachment for your convenience.

Reviewer 2 Report
The article “Investigation of an Impedancemetric LaSrMnO3-Au/Y2O3-ZrO2-Al2O3 Composite NOx Sensor” reports the development of a NOx sensor based on a composite material based on La-based perovskite with a ternary mixed oxide. The article is well presented and the results are very promising to scientific community of gases sensors, then I recommend the publication of manuscript with major revisions.
This way, I have comments, questions and suggestions that can enhance the performance of manuscript that are enumerated below.
1. In the Introduction section, the second paragraph is confused (lines 49-69). It is not clear for me why the metal oxides can enhance the sensitivity of NOx gases. Explain in a scientific manner. In special, why the addition of Sr in the La-based perovskites can enhance the performance of sensitivity?
2. I think the Fig 6 and Table 1 are not necessary in the Experimental section.
3. The second figure that have been shown is called “Fig 6” in Experimental Section. Verify!
4. In the Experimental Section: How was the electrochemical measurements made? I missed the details of experiments in all sections.
5. In the section 3.1.2, the authors say the electrolyte has cubic phase with trace amounts of monoclinical and tetragonal phases. What are the advantages and disadvantages of each phase in the final material?
6. In line 361 the authors say: Also, Sr2+ ion doping elevates the conduction pathway via oxygen species near the triple phase boundary. I think the authors need to explain better this sentence.
7. In Fig 7, the authors don’t explain the Figure completely. Why the temperature has been influenced the angular phase?
8. In the section 3.2.5 (Figure 9): I missed the linear equations and the R² of both FZC composite and FZC sensor. The angular coefficient may be attributed to the sensitivity of the sensor and thus the authors can show about the most sensitivity of FZC composite sensor than FZC sensor.
9. The electron transfer rate (k0) is shown in a wrong manner. The ‘k’ is lower case, not capital letter as shown in the manuscript.
10. I think the manuscript has so many figures in the main article. Some of Figures should be transferred to SI material.
11. The sections 3.2.3 and 3.2.4 the authors have demonstrated some impedance principles, that can be moved to Supplementary Material. I think the explanation is great, but these explanations have in so many books of Electrochemical Impedance Spectroscopy.
12. In the Reference Section: verify some inconsistences in formatting references and try to obtain more actual references of NOx gas sensors.
Author Response
We thank the reviewer for the work, time, and valuable comments and suggestions, drawing our attention to improve the manuscript. The revised manuscript has been prepared by considering the reviewer's recommendations, and the changes have been highlighted in yellow in the revised manuscript. Please find the point-by-point responses in the attachment to the reviewer's comments.

Round 2
Reviewer 2 Report
The authors made the suggested changes, improving the quality of the work. In my opinion, the manuscript can be accepted for publication.